# Genital Folliculosebaceous Cystic Hamartoma: A Case Report and Concise Review of the Literature

**Maged Daruish** [1,*] and **Mona Abdel-Halim Ibrahim** [2]

1   Department of Dermatopathology, St John's Institute of Dermatology,
    Guy's and St Thomas' NHS Foundation Trust, London SE1 7EH, UK
2   Department of Dermatology, Ain Shams University, Cairo 11517, Egypt; ibrahim.mona2010@gmail.com
*   Correspondence: maged.daruish@gstt.nhs.uk

**Abstract:** Folliculosebaceous cystic hamartoma (FSCH) is an uncommon hamartoma that usually presents on the central face area of adults as an asymptomatic, solitary dome-shaped or pedunculated papule. We report a case of a 35-year-old female who presented with six-months history of skin lesions on her labia majora. Histological findings included cystically dilated hair follicles with branching epithelial strands and interconnecting sebaceous gland consistent with the diagnosis of FSCH. The genital variant of FSCH was first described in 1998 and since then only six cases have been reported in the literature. We aim to increase awareness of this rare presentation due to the significant psychological implications and the risk of misdiagnosis.

**Keywords:** folliculosebaceous cystic hamartoma; sebaceous trichofolliculoma; genital hamartoma

## 1. Introduction

Folliculosebaceous cystic hamartoma (FSCH) is a hamartoma composed of follicular, sebaceous and mesenchymal elements. It usually presents as an asymptomatic, solitary dome-shaped or pedunculated papule or nodule on the head in adult patients, with predilection for the central face and nose [1]. Here, we report a case of the rare genital variant of FSCH.

## 2. Detailed Case Description

A 35-year-old female presented with multiple asymptomatic papules and nodules on the labia majora. The lesions were of 6 months duration and gradually increased in number and size. They were rounded, skin-colored, pedunculated, firm, had a smooth surface and ranged from 0.5–2 cm in diameter. The patient was concerned about having viral warts. Her husband did not have history of similar lesions. The clinical differential diagnosis included condyloma lata, condyloma accuminata, fibrolipomas, and adnexal tumors. Excision of 3 lesions were performed and histological examination revealed multiple cystically dilated and distorted hair follicles surrounded by thin anastomosing epithelial strands. Sebaceous glands were seen communicating with the cyst through small ductules. The surrounding stroma was composed of highly vascular, dense collagenous tissue and was separated from the surrounding adjacent uninvolved dermis by prominent clefting. A granulomatous inflammation was also noted (Figures 1–3). The features were found to be consistent with the diagnosis of genital folliculosebaceous cystic hamartoma.

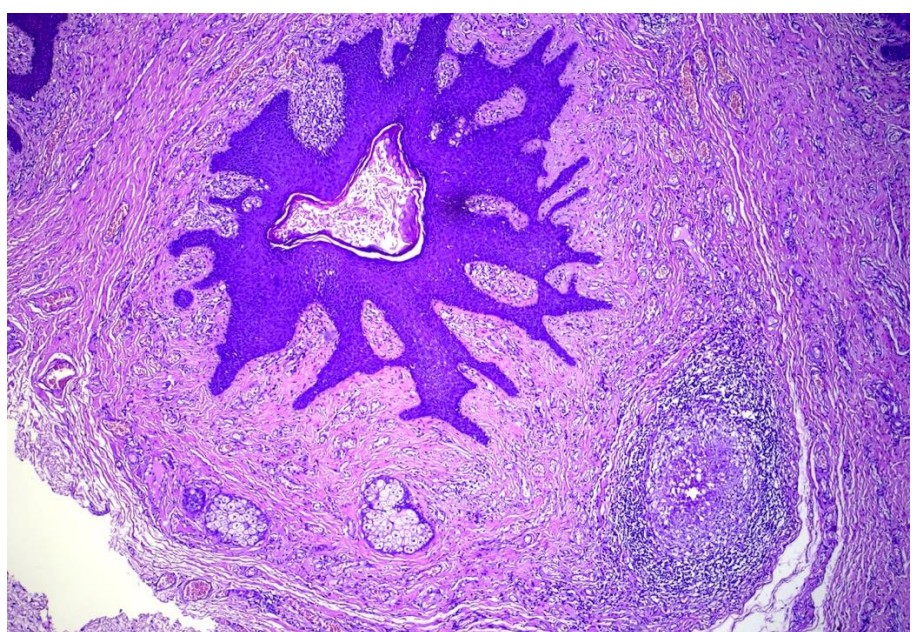

**Figure 1.** Cystically dilated hair follicle in the background of fibro-collagenous stroma. A secondary foreign body giant cell granulomatous reaction is also present. Hematoxylin and eosin, (H&E) ×100.

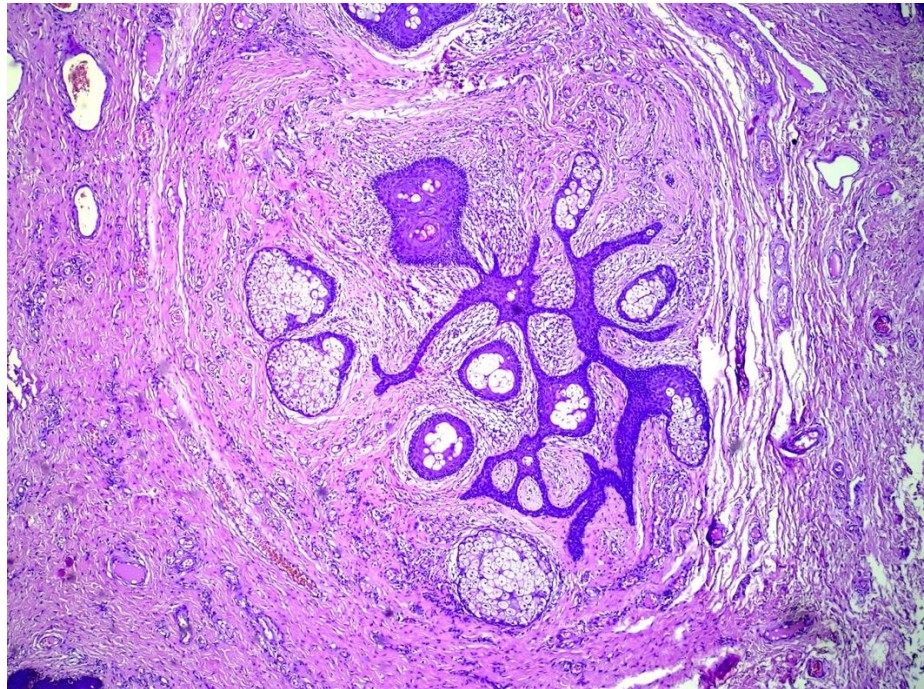

**Figure 2.** Anastomosing epithelial strands with multiple radiating sebaceous glands. Artefactual retraction can be seen between the mesenchymal component and surrounding uninvolved dermis. (H&E) ×100.

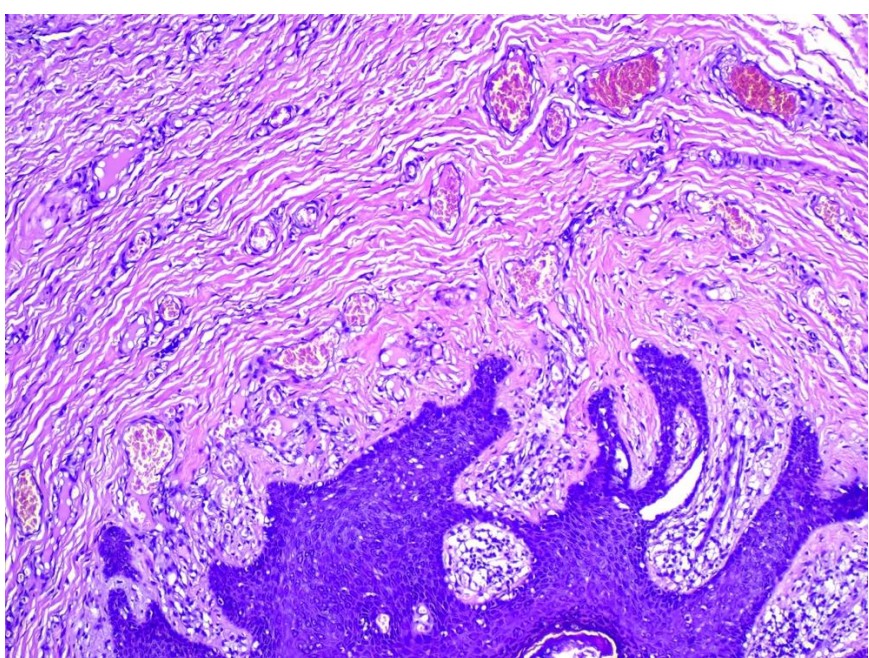

**Figure 3.** Stroma is collagenous and associated with numerous small sized blood vessels. (H&E) ×200.

## 3. Discussion

FSCH was first described as a distinct entity by Kimura et al. in a series of five cases in 1991 [2]. The main diagnostic criteria on histological examination include presence of an epithelial component in the form of an infundibular cystic structure with radiating connected sebaceous lobules, and a mesenchymal element composed of collagenous stroma, adipocytes, and small blood vessels [3].

Bolognia and Longley described the genital variant of FSCH in 1998 [4]. The authors argued that a case of scrotal sebaceous trichofolliculoma (STF) published in 1991 by Nomura and Hata [5] in fact represented FSCH. Further six cases were reported afterwards in the literature [6–11]. Genital FSCH mostly manifests as single nodule or multinodular lesion on the labia majora in young to middle-aged females. Clinical characteristics from reported cases are summarized in (Table 1).

FSCH does not have distinguishing features clinically and can be easily mistaken for neurofibroma, lipoma, intradermal nevus, and sebaceous hyperplasia [3]. Viral warts would be an important differential diagnosis in the context of genital lesions due to the potential misdiagnosis and the psychological distress it can cause to the patients. However, histological examination would be sufficient to differentiate FSCH [3].

The main histological differential diagnosis tends to be STF, which presents as a depressed lesion clinically. It tends to show hair shafts within the dilated hair follicle, displays branching finger-like smaller follicles, and a more frequent connection to the epidermis. It also lacks the characteristic stroma of FSCH [12]. Due to the overlapping features reported in some lesions, it is hypothesized that FSCH may represent a late stage in development of STF. Both share similar pathogenesis background in the form of CK15-positive hair follicle stem cells activation [13,14].

Treatment in most cases has been surgical excision [11]. Successful combination of $CO_2$ laser and acitretin has been described in one patient with a giant lesion [7].

**Table 1.** Clinical features of genital FSCH cases reported in the literature.

| Author | Age (Years) | Sex | Size | Site | Duration | Clinical Description | Notes |
|---|---|---|---|---|---|---|---|
| Bolognia and Longley [4] | 34 | F | 4 cm | Left labia majora | Not available | Skin-coloured multinodular plaque | |
| Hamada et al. [6] | 40 | F | 5 cm | Pubis | 13 years | Three skin-coloured aggregated nodules | |
| Brucher et al. [7] | 74 | M | 23 × 16 cm | Whole scrotum | Since birth | Giant multinodular skin-coloured to yellowish lesion | |
| Park et al. [8] | 28 | F | Not available | Labia majora | Not available | Not available | Full article is not available in English |
| Wu et al. [9] | 28 | F | 6 × 2 × 1.5 cm (left) and 5.5 × 2.5 × 2 cm (right) | Bilateral labia majora | More than 10 years | Skin-coloured polypoid and peduncu-lated hypertrophy | Associated with fibrous dysplasia syndrome |
| Alegría-Land et al. [10] | 45 | F | 2 cm | Middle of right labia majora | Long-term (patient uncertain) | Skin-coloured nodule | |
| Khan et al. [11] | 29 | F | 5 × 3 cm | Right labia majora | 3 months | Giant papu-lonodular lesion | |

## 4. Conclusions

To the best of our knowledge, we report the 8th case of genital FSCH. Awareness of this rare diagnosis is important due to the associated psychological distress, cosmetic and sexual issues, and anxiety of an HPV diagnosis.

**Author Contributions:** Writing—original draft preparation, M.D.; writing—review and editing, M.D. and M.A.-H.I.; supervision, M.A.-H.I. All authors have read and agreed to the published version of the manuscript.

**Funding:** This research received no external funding.

**Institutional Review Board Statement:** Not applicable.

**Informed Consent Statement:** Written informed consent has been obtained from the patient to publish this paper.

**Conflicts of Interest:** The authors declare no conflict of interest.

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
