# Peer review of "Genital Folliculosebaceous Cystic Hamartoma: A Case Report and Concise Review of the Literature"

_dermatopathology, doi:10.3390/dermatopathology9030032_

Round 1
Reviewer 1 Report
- It describes a rare entity, potentially increasing awareness of it;
- It is well written and structured;
- The photomicrographs are representative of the main features of the entity and correspond to what is described in the text and legends.
- It has educational value, because it includes a literature review summarizing the features of previously described cases, highlighting similarities and differences between them (the table provided is also helpful for this).
The only main weakness I find is the lack of a clinical picture. Although not absolutely essential, it would improve the paper.
Author Response
We would like to thank the reviewer for his/her encouraging comments.
We entirely agree that clinical images would have been a major addition to the manuscript. Unfortunately, these were not taken at the time, and it is unlikely that would be able to obtain them at the current stage.
Reviewer 2 Report
The reviewer wishes to thank the editor and the authors for the opportunity to review this well-written and well-illustrated manuscript. This article seeks to add an additional case of genital folliculo-sebaceous cystic hamartoma to the literature in addition to raising awareness of this entity by also summarizing the pertinent clinical characteristics of the seven prior cases of genital FSCH reported in the literature.
Although this is a well described entity in the head and neck region, the authors seek to describe this case as clinically, in this particular anatomic location and in the setting of a married individual, it was highly concerning for an HPV-induced lesion. Given the stigma of HPV-induced lesions and the potential implications associated with new-onset lesions occurring during marriage, the reviewer agrees with the authors that this is important diagnosis to recognize in this location not only for dermatopathologists but also for clinicians.
Given that time course in this case was relatively short (6 months), the manuscript may be more attractive if time courses of the other lesions in the literature (when available) were added to Table 1 as a separate column. Additionally, it may of interest to describe if any of the reported cases other than the case described by Wu et al have been seen in association with a syndrome.
Author Response
We would like to thank the reviewer for his/her encouraging comments.
-We will add the suggested column with the duration of cases available in the literature. Interestingly, only one case had a short clinical course. The others were either not available or long-standing.
-To the best of our knowledge, there are no other available cases in the literature at this moment with associated syndromes. However, we will do another search before submitting the revision to be sure we are not missing anything.